# A Text-Mining Analysis of Research Trends in Animal-Assisted Therapy

**DOI:** 10.3390/ani13193133

**Published:** 2023-10-07

**Authors:** Shin-Ja Lee, Geun-Hyeon Kim, Yea-Hwang Moon, Sung-Sill Lee

**Affiliations:** 1University-Centered Labs, Institute of Agriculture and Life Science, Gyeongsang National University, Gyeongsangnam-do, Jinju-si 52828, Republic of Korea; tlswk1000@hanmail.net; 2Jeonju City Council Legislative Policy Division, Wansangu Nosonggwangjang-ro 10, Jeollabuk-do, Jeonju-si 54994, Republic of Korea; kimgh0307@korea.kr; 3Division of Animal Bioscience and Integrated Biotechnology, Gyeongsang National University, Gyeongsangnam-do, Jinju-si 52725, Republic of Korea; yhmoon@gnu.ac.kr; 4Division of Applied Life Science (BK21), Gyeongsang National University, Gyeongsangnam-do, Jinju-si 52828, Republic of Korea

**Keywords:** animal-assisted therapy, text mining, big data

## Abstract

**Simple Summary:**

Animal-assisted therapy (AAT) has recently gained increasing attention in various disciplines. However, understanding of the specific fields and topics being studied in this area remains limited, with most studies conducted using experimental approaches. Therefore, this study determined trends in AAT research based on an analysis of studies published via PubMed (*n* = 776 studies). Python programming was used for the analysis, employing techniques such as word cloud, n-gram, two-way word tree, and CONCOR (convergence of iteration correlation) analyses.

**Abstract:**

Text-mining techniques were used to provide basic data to related policy stakeholders and academic researchers by collecting and analyzing research trends related to animal-mediated healing in a short time. A total of 776 studies were collected using the keyword “animal-assisted therapy” (AAT) in the search engine PubMed, which covers a wide range of topics related to health sciences, biomedical research, and health psychology. Four analysis methods were employed. “Dog” was the most commonly utilized animal in AAT. This study also identified individuals with autism spectrum disorder and post-traumatic stress disorder as the primary research participants. Finally, the terms “health care” and “blood pressure” were identified, indicating that AAT has a positive impact on improving blood pressure and enhancing heart rate. These findings demonstrate that AAT research is being actively pursued in various fields, such as social sciences, medicine, and psychology.

## 1. Introduction

Recently, keywords related to healing, such as stress and depression, have rapidly become more common in the literature, leading to increased interest in and demand for animal-assisted interventions, including healing agriculture. The introduction of animal-assisted therapy (AAT) can be traced back to 1962 when Boris Levinson, an American child psychologist, published a study, “The dog as co-therapist”, which shed light on AAT [1]. In the 1970s, the use of animals in psychological healing programs for patients with mental illness who had lost self-control owing to their experiences in World War II was a starting point [2]. Since then, various programs using dogs, cats, and other animals have been implemented in schools, hospitals, and prisons. In the 1980s, research on AAT began in earnest, and studies reported that companion animals had a highly positive impact on human physical and mental health [3,4].

AAT is being integrated into various practical and academic fields. Notably, randomized controlled trials have been conducted. The animals used in these trials include dogs, cats, dolphins, birds, cows, rabbits, guinea pigs, and penguins. The results indicated that AAT was effective for people who liked animals and had mental and behavioral disorders, such as depression and schizophrenia. However, AAT was less effective for persons without an established fondness for animals [5]. Additionally, for individuals who have experienced trauma from war, sexual abuse, or other traumatic stressors and do not respond to medication, a new treatment method is needed.

AAT has been suggested to be helpful for treating post-traumatic stress disorder (PTSD), and experimental studies using dogs have indicated that AAT plays a role in preventing suicide in patients with PTSD [6]. In addition, cognitive dysfunction dementia is a chronic disease that affects social behavior and daily behavior performance, and there is no effective treatment to prevent the progression of the disease thus far. Most treatments have only employed symptomatic therapy, but animal-assisted treatment could be effective for patients with dementia. Since traditional healthcare still often fails to provide solutions for patients, other methods should be devised, and animal-assisted treatments for symptom management can greatly help patients with pain, pain relief, and improving quality of life [7].

During the COVID-19 pandemic, AAT improved the social functioning and quality of life of patients with schizophrenia. A study was conducted at six mental rehabilitation institutions in Taiwan in which participants watched short films about animals once a week for 60 min over a period of 12 weeks. AAT had a positive impact on schizophrenia; thus, AAT is recommended to promote community-based rehabilitation [8]. Both patients and nurses experienced prolonged burnout owing to long working hours during the COVID-19 pandemic, and AAT was identified to improve employee well-being [9]. Additionally, with the advancement of artificial intelligence in the era of the Fourth Industrial Revolution, the impact of robotic and live dogs on AAT has been examined. Both live and robotic dogs reduced loneliness and attachment in elderly patients but could not fully replace human interaction [10].

Most animal-mediated healing-related studies have been performed in the medical field; thus, more psychological research needs to be conducted. The range of studies is expanding and diversifying; thus, research trends should be examined through text-mining analysis techniques. Recently, text-mining techniques were applied to the agricultural [11], economic [12], and medical fields [13], but research insufficient in animal-mediated healing is insufficient. AAT has been studied in various fields, underscoring the need for animals as an element of the healing process. Animal-mediated treatment has preventive and special-purpose subjects, and it is not necessary to distinguish the effects of animal-mediated treatment using animal resources just because the scope of the thesis varies depending on the purpose.

The significance of this study lies in its collection and analysis of data from studies related to AAT. Analyzing ample data in detail is not only time-consuming but also requires familiarity with the content. Python was used to analyze the research trends and the focus of the keywords in the AAT literature to identify areas that lack exploration. Based on these results, suggestions are provided to practitioners in the field of AAT and offer academic insights to researchers. In conclusion, words related to medical care were derived from the results obtained using text -mining techniques This is not to conclude that medical services should be focused on animal-mediated treatment but to identify research trends in animal-mediated treatment.

## 2. Materials and Methods

### 2.1. Research Method

This study examined research trends related to AAT. To access a wide range of health and welfare fields, including life sciences, biomedical research, and health psychology, this study primarily utilized the MEDLINE database, which is accessible through the PubMed search engine. A total of 776 studies were collected by searching for the keyword “Animal-Assisted Therapy”, while the titles and abstract keywords were retrieved. To refine the collected keywords, Python’s “re” package was used to remove special characters and text-stop words. Subsequently, a custom spelling checker (from Busan National University) was used for a primary refinement. Nouns, adjectives, and verbs were extracted using the Konlpy package, while word frequencies were calculated using the counter function in the collection module. Words with the top 100 frequencies were subjected to further refinement and stop-word processing, resulting in three or more occurrences.

Four analysis methods were employed with refined words as the focus. The analysis techniques included word cloud, n-gram, two-way word tree, and CONCOR (convergence of iteration correlation) analyses. The analysis process (Figure 1) and the concepts and definitions of each analysis method are summarized in the following subsection.

### 2.2. Analysis Techniques

The first technique used in this study was a word cloud analysis, which is a widely used method to visually represent words. A word cloud analyzes the frequency of words used in text and visualizes them based on their size and intensity [14]. N-gram is a technique that allowed us to examine the strength of the connections between words; depending on the value of N, it can be expressed as a unigram, bigram, trigram, tetragram, and so on. Bigrams were used in the analysis. To use the bigram, the words must be paired in groups of two, which was achieved using the zip function. The strength of the connections between the paired words was analyzed using value_counts. The DiGraph function from the Python network visualization module was used to visualize this as a network [15]. A two-way word tree is a technique that allowed us to determine the importance of keywords in the relationships between words [16]. Finally, a CONCOR analysis was conducted via Python to calculate the frequency of the simultaneous appearance of the top 100 words in each paragraph and converted the word list into a one-mode symmetric matrix (word × word) via the Ucinet 6.0 ver. program. The NetDraw program then analyzed the correlation and visualized and clustered the network between words [17].

## 3. Results

### 3.1. Word Cloud Analysis

According to the word cloud analysis results (Table 1, Figure 2), the term “dog (810)” had the highest frequency among the keywords corresponding to the animals used in healing therapy. Cheon and Jung stated that dogs are highly sociable and easily adaptable to all age groups, which has led to most research being conducted with them. Next, the term “horse (68)” was identified to be prominent as well. Horses have been used in physical therapy to improve body balance and movement; however, more recently, they have been used in educational and psychological therapy [18]. Rothe et al. noted that horses are intuitive and sensitive animals that can sense human moods and respond immediately, thereby enhancing trust, respect, self-confidence, and communication abilities in children with conditions such as autism spectrum disorder (ASD), PTSD, and disabilities. The frequency of references to dogs and horses was high in the studies related to AAT, indicating the need for research and experiments involving other animals to increase the reliability and accuracy of AAT [19].

Keywords representing participants such as “depression (88)”, “PTSD (84)”, “disorder (227)”, “dementia (68)”, “stress (202)”, “anxiety (158)”, “autism (97)”, and “children (52)” were also prominent. AAT has been studied and applied in individuals experiencing depression, anxiety, and stress. Numerous studies have focused on individuals with conditions such as dementia, ASD (especially children), and PTSD. Animal-based therapies are effective when conventional drug-based treatments are challenging for individuals with mental health issues [6].

Finally, terms indicating the effects of AAT, such as “effect (335)”, “program (318)”, “benefit (167)”, “improve (152)”, “increase (147)”, “reduce (121)”, “aim (110)”, “rehabilitation (71)”, and “decrease (56)”, were identified. AAT programs were confirmed to contribute to improving physical, emotional, and cognitive functions and enhancing quality of life. In addition to promoting mental stability, AAT has shown benefits such as pain reduction and improvements in social and expressive skills [20]. It is a useful tool for reducing attention-deficit/hyperactivity disorder (ADHD) in children [21]. The use of AAT in various fields, such as social sciences and medicine, has been acknowledged, and its positive effects have been diverse.

### 3.2. N-Gram Analysis

The strong association between the terms (Table 2, Figure 3) “therapy dog” and “service dog” indicates that the use of dogs in AAT is actively being pursued. Further, the presence of terms such as “control group”, “randomized control”, “assist intervention”, and “pilot study” suggests that experiments targeting specific groups, such as older persons and children with autism, have been conducted using randomized control groups. Additionally, terms such as “quality of life” and “post-traumatic stress” indicate that survey-based social science methods have been employed to assess improvements in quality of life and stress reduction. Moreover, medical research has been conducted using objective measures, such as heart rate and blood pressure, highlighting the use of medical approaches. Kim argued for the need to enhance the reliability and scientific evidence of AAT by conducting research with quantifiable indicators, such as reductions in antidepressant medication dosage, decreased suicide rates, and pain relief among patients receiving therapy [22]. Although the frequencies of quantifiable indicators may not have been high in this study, utilizing such indicators in research could increase their credibility.

Finally, the appearance of terms such as “nurse home” and “dog visit” indicates that AAT often involves nurses and dogs visiting the homes of individuals with limited mobility. In Japan, the Animal Hospital Welfare Association instigated the Human-Animal Interaction Partner Program in 1986, which recognizes volunteer activities involving animals visiting various facilities and became a corporation under the jurisdiction of the Ministry of Health, Labour and Welfare. The Pet Partner Program involves well-trained animals visiting hospitals and schools to engage in activities that promote the rehabilitation of individuals in good health. Following visits by therapy dogs, patients have shown increased interactions with hospital staff [23]. While these studies support the basis of this study, it is necessary to examine the results of AAT conducted after the COVID-19 pandemic and to understand the impact of remote interactions on AAT professionals, animals, and patients.

### 3.3. Two-Way Word Tree

The advantage of the two-way word tree analysis technique is that it is possible to grasp not only the front and back of the word to be identified but also the extended range of words. Figure 4a focuses on “assist” and has high references to the effects and improvements of animal-mediated healing such as “efficiency, therapy, review, effect, and improve”. Figure 4b focuses on “benefit” and suggests that health benefits can be received through pets such as “health, pet, therapy, research, treatment, and risk”. Figure 4c focuses on “effect” and shows that there are many studies on animal-mediated healing effects related to dogs such as “study, evaluate, reduce, anxiety, dog, and therapy”. Figure 4d focuses on “stress”, and words that represent negative emotions can be reduced through animal-mediated healing such as “reduce, post, report, effect, disorder, PTSD, pain, improve, and anxiety”. Figure 4e focuses on “automatism”, and many animal-mediated healing studies addressed children with autism and revealed words such as “assist, youth, dog, spectrum, disorder, assistance, and dog”. Lastly, Figure 4f focuses on “PTSD”, and many studies addressed healing and symptoms using dogs for post-traumatic disorders such as “treatment, stress, post, treatment, disorder, symptom, study, service, and dog”.

### 3.4. CONCOR Analysis

The size of each node indicates the frequency of the keyword, and the thicker the connecting line between the nodes is, the higher the connection [24]. Looking at Figure 5, the connection strength of cluster 1 was 29.091 and the connection strength of cluster 8 was 6.333, showing the lowest cluster. The R-squared value is 0.061, showing a relatively high explanatory power. It is a concept representing the degree of a relationship between nodes, and it is in the range between 0 and 1. If the explanatory power is 0.5 or more, the connection strength is deemed very high [24].

In the CONCOR analysis (Figure 6), the word cluster “Rehabilitation” consisted of words such as “program”, “condition”, “time”, “stress”, and “reduce”. This suggests that AAT primarily focuses on rehabilitation programs and the effects and time required to achieve them. The importance of rehabilitation in AAT can be inferred from these findings.

The word cluster “Benefit” included words such as “help”, “develop”, “welfare”, “health”, “work”, and “staff”. This indicates that AAT provides welfare benefits to professionals working in this field and leads to health improvements. It is evident that AAT is not only beneficial for individuals with physical or mental discomfort or social vulnerability but also for employees experiencing stress in their workplace as they receive welfare benefits and related support.

The word cluster “Companion” comprised words such as “visit”, “support”, “family”, and “assistance”. This indicates that AAT involves professionals and animals visiting the homes of older or disabled individuals with mobility limitations to provide them with assistance. This indicates a shift in AAT practices from patients visiting specialized AAT centers to experts visiting patients’ homes to deliver services.

The word cluster “Pet” was associated with words such as “life, attention, base, review, improve, approach, home, process, interaction, dementia, quality, disease, horse, and care”. This reveals that AAT is being explored in the literature for treating patients with dementia and the processes involved. This confirms that research on AAT is being conducted across various fields, including the social sciences and medicine.

The word cluster “Research literature” included words such as “paper, literature, evidence, article, term, practice, and animal, role”. This indicates that a significant amount of research has been conducted on the treatment of patients with dementia and the processes involved in AAT. This highlights that research is being actively conducted on AAT in various disciplines.

The word cluster “Effectiveness” consisted of words such as “depression, show, treatment, pain, effectiveness, AAT, group, function, session, randomize, change, efficacy, trial, level, test, control, and age”. This indicates that AAT has been effective at reducing depression and alleviating pain. This indicates that research is being conducted to demonstrate the effectiveness of AAT, using multiple testing processes.

Finally, the clusters “Service center (train, PTSD, presence, analysis, impact, report, relate, dog, service, center)” and “Object to be healed (child, spectrum, parent, autism, communication)” revealed the diverse range of individuals benefiting from AAT, including children with autism and their parents. Service centers aim to provide therapy to these individuals. The parents of children with physical or mental discomfort are also considered recipients of AAT. The range of target recipients is expanding, indicating increasing recognition of the importance of AAT. In future research, it is anticipated that AAT studies targeting workers from various fields (e.g., construction workers and night-shift workers) will further underscore the significance of AAT.

## 4. Discussion

Research on AAT has primarily focused on dogs and horses. Cheon and Jung’s review reported that dog-assisted therapy accounted for 48% of the studies, followed by horse-assisted therapy (34%). These studies predominantly targeted children and adolescents, whereas fish and birds were more commonly used with older persons [18]. Kim et al. examined 68 AAT studies, with 50 focusing on dogs and 13 on horses [25]. This finding suggests the need to explore the use of various animals for AAT rather than remaining limited to dogs or horses. Further research should be conducted to identify differences from the literature, explore trends in AAT research in other countries, and delve into the specifics of these findings.

Terms indicating the target population in need of AAT emerged prominently, such as “disorder”, “stress”, and “anxiety”, which refer to individuals experiencing emotional distress. Additionally, specific groups were identified through words such as “autism”, “PTSD”, “parents”, and “children”. Granger and Kogan and O’Haire confirmed the effectiveness of AAT at enhancing self-management skills [26,27] and reducing stress, whereas Barker et al. highlighted its positive impact on emotional regulation, noting that most experimental subjects of AAT research were young children [28]. Various fields have been explored, including ADHD, ASD, and child-abuse-related PTSD. According to Becker et al. and Rothe et al., children with attention deficit/hyperactivity disorder experience behavioral improvements and enhanced socialization skills through AAT [19,29]. Gillett and Weldrick noted its usage primarily among traumatized military personnel but highlighted its recent application in diverse populations [30]. Interaction with animals can reduce loneliness and isolation, increase social interactions, and facilitate reintegration into daily life [31].

According to Kang and Oh, AAT reduces parental anxiety and improves parenting self-efficacy by providing mothers with the necessary understanding and skills to interact with their children with disabilities [32]. Further, Florence and Eddins-Folensbee supported the notion that therapy dogs and related activities brought joy and psychological stability to mothers, which positively impacted their children [33].

The predominance of specific target populations for AAT was confirmed, highlighting the need for future studies to explore its application to diverse subjects such as stressed working professionals or individuals involved in pet-related occupations. Various terms indicating the effects of AAT were identified, such as “quality of life”, “healthcare”, “blood pressure”, and “heart rate”, which highlighted the positive effects of AAT in improving quality of life, promoting health, lowering blood pressure, and stabilizing heart rate. Research on quality of life falls within the social sciences, whereas areas such as blood pressure and heart rate belong to the medical field, emphasizing the need for an interdisciplinary analysis that combines findings from both fields. By integrating multiple disciplines into the analysis, the reliability and validity of this research will be enhanced, resulting in higher-quality studies.

## 5. Conclusions

This study examined research trends in the literature related to AAT, using text-mining techniques. It identified the main topics and subjects in this field. Four techniques, word cloud, n-gram, two-way word tree, and CONCOR analyses, were utilized via Python. By collecting and refining a vast amount of literature, this study expedited the process and objectively transformed the data into a usable format.

This study had some limitations. First, the authors’ subjectivity could have influenced the results during the process of refining a vast amount of data, leading to a potential lack of objectivity. Second, because this study focused primarily on the keyword AAT in the PubMed search engine, it could be difficult to generalize the results.

In future research, other programs or tools could help refine the data and promote objectivity in word analysis. Additionally, future research should collect data from various search engines that extensively cover AAT studies and conduct comparative analyses.

## Figures and Tables

**Figure 1 animals-13-03133-f001:**
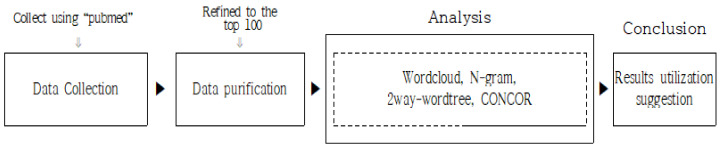
Research design.

**Figure 2 animals-13-03133-f002:**
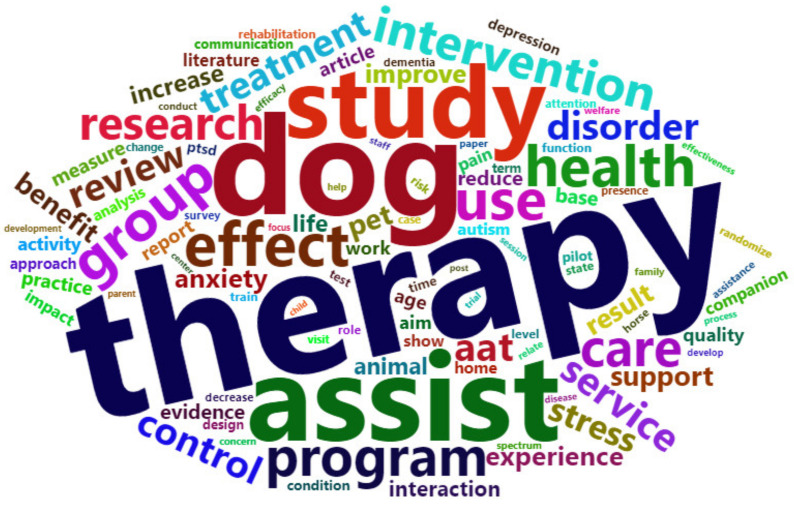
Word cloud visualization.

**Figure 3 animals-13-03133-f003:**
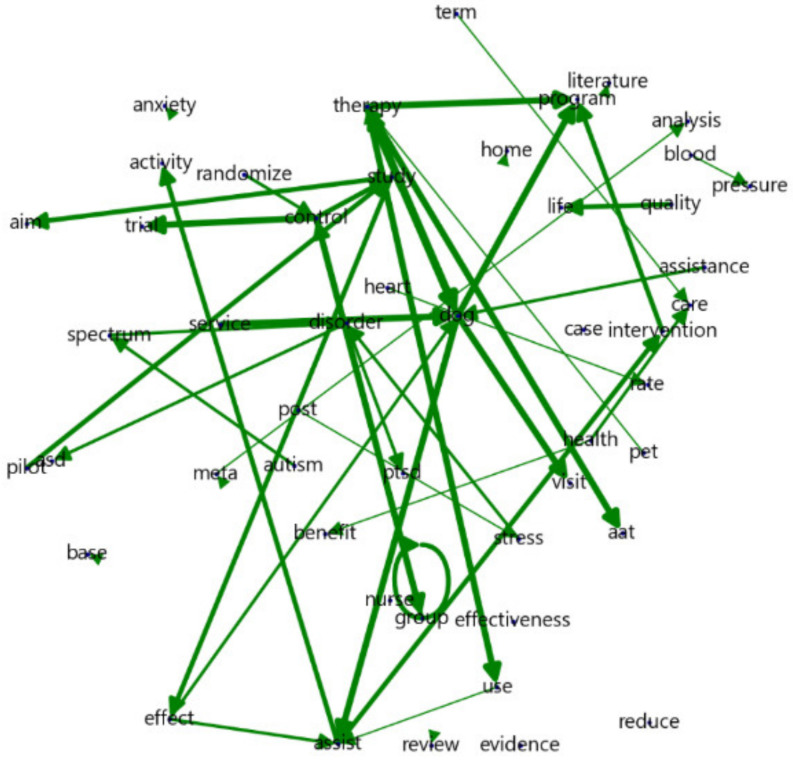
N-gram visualization.

**Figure 4 animals-13-03133-f004:**
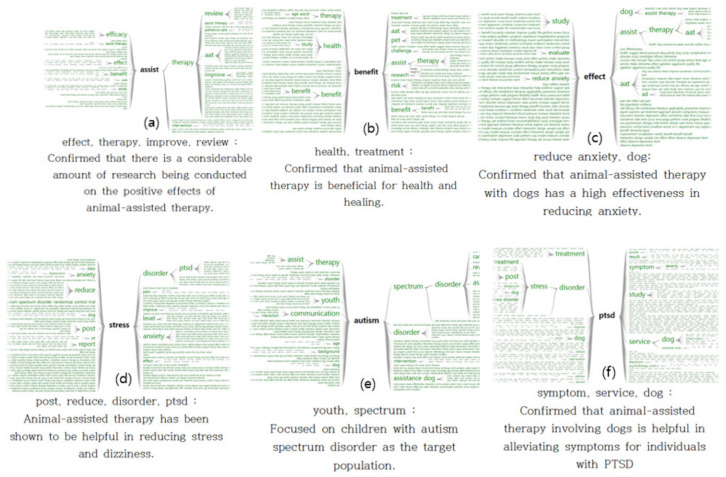
Two-way word tree visualization: (**a**) effect, therapy, improve, and review confirmed that there is a considerable amount of research being conducted on the positive effects of animal-assisted therapy; (**b**) health, treatment, benefit, and risk confirmed that animal-assisted therapy is beneficial for health and healing; (**c**) reduce, anxiety, dog, and aat confirmed that animal-assisted therapy with dogs has a high effectiveness in reducing anxiety; (**d**) post, reduce, disorder, and PTSD indicated that animal-assisted therapy has been shown to be helpful in reducing stress and dizziness; (**e**) communication, youth, and spectrum indicate that on children with autism spectrum disorder were focused on as the target population; (**f**) symptom, service, dog, and disorder confirmed that animal-assisted therapy involving dogs is helpful in alleviating symptoms for individuals with PTSD.

**Figure 5 animals-13-03133-f005:**
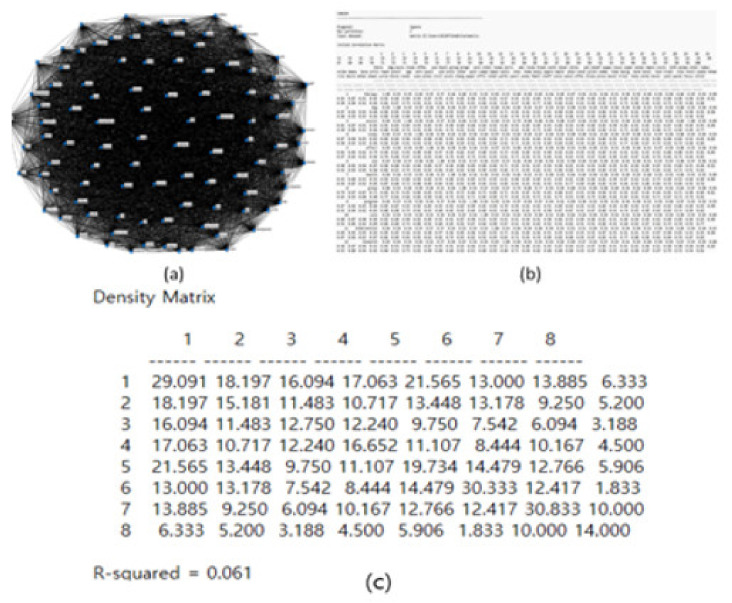
CONCOR analysis calculation derivation process.

**Figure 6 animals-13-03133-f006:**
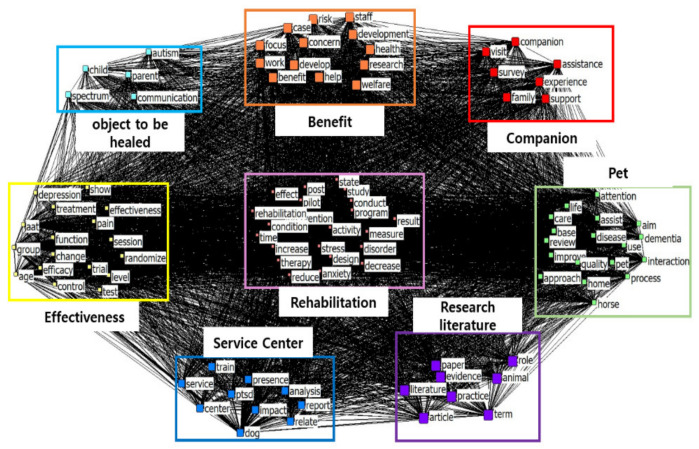
CONCOR Visualization.

**Table 1 animals-13-03133-t001:** Word cloud analysis results.

Rank	Keyword	Frequency	Rank	Keyword	Frequency
1	Therapy	1087	51	Approach	89
2	Dog	810	52	Depression	88
3	Assist	724	53	PTSD	84
4	Study	587	54	Condition	82
5	Effect	355	55	Pilot	82
6	Use	340	56	Communication	79
7	Health	339	57	Time	79
8	Group	339	58	Design	78
9	Program	318	59	Term	77
10	Care	305	60	Level	75
11	Intervention	290	61	Test	75
12	Research	275	62	Train	74
13	Service	240	63	Role	74
14	AAT	238	64	Function	73
15	Review	231	65	State	72
16	Treatment	229	66	Rehabilitation	71
17	Control	228	67	Risk	70
18	Disorder	227	68	Decrease	69
19	Stress	202	69	Attention	69
20	Pet	169	70	Dementia	68
21	Benefit	167	71	Survey	68
22	Support	162	72	Horse	68
23	Result	162	73	Randomize	67
24	Experience	158	74	Case	66
25	Anxiety	158	75	Presence	66
26	Improve	152	76	Visit	65
27	Increase	147	77	Assistance	65
28	Life	139	78	Change	64
29	Animal	136	79	Paper	61
30	Interaction	128	80	Efficacy	61
31	Reduce	121	81	Relate	60
32	Evidence	119	82	Welfare	60
33	Measure	117	83	Spectrum	60
34	Base	117	84	Conduct	60
35	Article	116	85	Family	59
36	Report	115	86	Staff	58
37	Practice	112	87	Concern	58
38	Age	112	88	Session	57
39	Work	111	89	Effectiveness	57
40	Quality	110	90	Disease	56
41	Aim	110	91	Process	56
42	Activity	107	92	Develop	56
43	Literature	103	93	Trial	56
44	Pain	103	94	Help	55
45	Companion	99	95	Center	54
46	Impact	98	96	Development	54
47	Autism	97	97	Post	54
48	Show	97	98	Parent	53
49	Home	94	99	Focus	52
50	Analysis	91	100	Child	52

**Table 2 animals-13-03133-t002:** N-gram analysis results.

N	Keyword A	Keyword B	Strength	N	Keyword A	Keyword B	Strength
1	Assist	Therapy	445	51	benefit	assist	17
2	Therapy	Dog	147	52	reduce	stress	17
3	Service	Dog	129	53	dog	dog	16
4	Therapy	AAT	114	54	dog	train	16
5	Control	Group	77	55	train	animal	16
6	Assist	Intervention	70	56	companion	dog	16
7	Quality	Life	70	57	ptsd	service	16
8	Pilot	Study	61	58	therapy	assist	16
9	Effect	Assist	60	59	care	home	16
10	Stress	Disorder	58	60	therapy	treatment	15
11	Autism	Spectrum	55	61	effect	aat	15
12	Randomize	Control	52	62	depression	anxiety	15
13	Spectrum	Disorder	51	63	brain	injury	15
14	Study	Effect	46	64	animal	welfare	15
15	Dog	Assist	44	65	effect	pet	15
16	Assistance	Dog	41	66	dog	stress	15
17	Therapy	Program	41	67	facility	dog	14
18	Disorder	PTSD	37	68	therapy	autism	14
19	Health	Care	35	69	anxiety	depression	14
20	Pet	Therapy	34	70	day	care	13
21	Control	Trial	33	71	effect	therapy	13
22	Heart	Rate	32	72	study	investigate	13
23	Assist	Activity	31	73	study	report	13
24	Aim	Study	30	74	stress	anxiety	13
25	Blood	Pressure	28	75	dog	reduce	13
26	Effect	Dog	28	76	purpose	study	13
27	Use	Assist	27	77	study	use	13
28	Health	Benefit	26	78	pain	management	13
29	Meta	Analysis	25	79	dog	handler	13
30	Post	Stress	24	80	dog	cat	13
31	Group	Group	24	81	efficacy	assist	12
32	Control	Study	24	82	pet	ownership	12
33	Dog	Program	23	83	pressure	heart	12
34	Term	Care	23	84	study	design	12
35	Review	Literature	23	85	study	assist	12
36	Study	Aim	22	86	review	review	12
37	Dog	Therapy	22	87	improve	health	12
38	Nurse	Home	21	88	group	therapy	12
39	Evidence	Base	21	89	therapy	therapy	12
40	Literature	Review	20	90	tf	cbt	12
41	Case	Study	20	91	presence	dog	12
42	Intervention	Program	20	92	study	study	12
43	Effectiveness	Assist	19	93	dog	use	12
44	Group	Control	19	94	intervention	group	12
45	Use	Therapy	19	95	assist	assist	11
46	Disorder	ASD	19	96	result	show	11
47	Dog	Visit	18	97	train	dog	11
48	Reduce	Anxiety	18	98	age	group	11
49	Review	Meta	17	99	dog	health	11
50	Therapy	Use	17	100	health	condition	11

## Data Availability

The data presented in this study are available on request from the corresponding author. The data are not publicly available due to privacy.

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
