# Peer review of "A Text-Mining Analysis of Research Trends in Animal-Assisted Therapy"

_animals, 2023, doi:10.3390/ani13193133_

Round 1

Reviewer 1 Report

The paper covers an area of great value and methods (Text mining) which are becoming more valuable to the systematic review process. However I feel the paper need to focus either on the value of text mining to systematic review using AAT as a model OR AAT as an intervention in healthcare. The scope of the paper cannot do both.

Abstract - Line 24 - AAT cannot be a solution to the problems, just an intervention that may aim to improve the outcomes. The aims should be more focussed. 

Introduction - I noticed at the start of the paper (line 44) ref 2 was used inappropriately. The paper referenced is not the study cited. A reference for the actual study should be cited. Line 48-58 the info provided whilst concerning and of relevance to the authors, is less important with regard to the focus of the paper. In addition the information provided to describe the areas AAT has been used, whilst interesting, was not described appropriately. More focus should have been given to why text mining per se was used and the aims of the research.

Author Response

Comments and Suggestions for Authors: The paper covers an area of great value and methods (Text mining) which are becoming more valuable to the systematic review process. However I feel the paper need to focus either on the value of text mining to systematic review using AAT as a model OR AAT as an intervention in healthcare. The scope of the paper cannot do both.

Point 1: Abstract - Line 24 - AAT cannot be a solution to the problems, just an intervention that may aim to improve the outcomes. The aims should be more focused.

Introduction - I noticed at the start of the paper (line 44) ref 2 was used inappropriately. The paper referenced is not the study cited.

Response 1: I have cited references that align with the content as pointed out by the reviewer. Thank you.

Point 2: A reference for the actual study should be cited. Line 48-58 the info provided whilst concerning and of relevance to the authors, is less important with regard to the focus of the paper.

Response 2: You have revised it as follows based on the author's feedback: "Text-mining techniques were used to provide basic data to related policy stakeholders and academic researchers by collecting and analyzing research trends related to animal-mediated healing in a short time."

Point 3: In addition the information provided to describe the areas AAT has been used, whilst interesting, was not described appropriately. More focus should have been given to why text mining per se was used and the aims of the research.

Response 3: Based on the reviewer's feedback, you have revised it as follows:

" In addition, cognitive dysfunction dementia is a chronic disease that affects social be-havior and daily behavior performance, and there is no effective treatment to prevent the progression of the disease so far. Most treatments have only employed symptomatic ther-apy, but animal assistance treatment could be effective for patients with dementia. Since traditional healthcare still often fails to provide solutions to patients, other methods should be devised, and animal treatment for symptom management can greatly help pa-tients with pain, pain relief, and quality of life improvement."

Point 4: Most animal-mediated healing-related studies have been performed in the medical field; thus, more psychological research needs to be conducted. The range of studies is expanding and diversifying; thus, research trends should be examined through text-mining analysis techniques. Recently, text-mining techniques have been applied to the agricultural, economic and medical fields, but research is insufficient in animal-mediated healing.

Response 4: Based on the reviewer's feedback, you have revised it as follows:

" It was found that text mining was used in various research fields, but ATT research pointed out that text mining utilization was insufficient. (Modified it like this)”.

Reviewer 2 Report

-a conclusion should be provided after comparing some results from the proposed methods.  (i.e. the two way word tree and the CONCOR analysis that have a common term “benefit”).

-more information is needed for the role of CONCOR analysis and whether  there is any advantage over the n-gram method.

Author Response

Point 1: a conclusion should be provided after comparing some results from the proposed methods.  (i.e. the two way word tree and the CONCOR analysis that have a common term “benefit”).

Point 2: more information is needed for the role of CONCOR analysis and whether there is any advantage over the n-gram method.

Response 1-2: Thank you for your guidance. I have made the necessary revisions based on the reviewer's feedback.

The advantage of the two-way word tree analysis technique is that it is possible to grasp not only the front and back of the word to be identified but also the extended range of words. 3(a) focuses on “assist” and has high references to the effects and improvements of animal-mediated healing such as “efficiency, therapy, review, effect, and improve.” 3(b) focuses on “benefit” and suggests that health benefits can be received through pets such as “health, pet, therapy, research, treatment, and risk.” 3(c) focuses on “effect” and shows that there are many studies on animal-mediated healing effects related to dogs such as “study, evaluate, reduce, anxiety, dog, and therapy.” 3(d) focuses on “stress” and words that represent negative emotions can be reduced through animal-mediated healing such as “reduce, post, report, effect, disorder, PTSD, pain, improve, and anxiety.” 3(e) focuses on “automatism” and many animal-mediated healing studies addressed children with autism and revealed words such as “assist, youth, dog, spectrum, disorder, assistance, and dog.” Lastly, 3(f) focuses on “PTSD” and many studies addressed healing and symptoms using dogs for post-traumatic disorders such as “treatment, stress, post, treat-ment, disorder, symptom, study, service, and dog.” (Modified it like this)

Reviewer 3 Report

1.) Is CONCUR an acronym? If so, it needs to be spelled out in first occurrence in Abstract line 20. CONCUR analysis is discussed in line 120 but no explanation or presentation of formula used to perform these calculations mentioned on lines 120 and 121. CONCUR Analysis is further discussed in section 3.4 but no formula(s) for the calculations discussed in line 120 are presented in this section 3.4.

2.) This paper presents visual presentations of the techniques of word cloud, n-gram, two-way word tree and CONCUR analysis but do not provide any of the calculations referred to in the descriptions of CONCUR Analysis as it does for the others as in sections 3.1, 3.2 and 3.3.

3.) Table 1 is used twice for "Wordcloud Analysis Results" on page 4 and also for N-gram Analysis Results on page 6 that instead should be Table 2. There are no figure numbers for the 6 figures on page 7 that maybe should be labeled as Figures 3(a), 3(b), 3(c), 3(d), 3(e) and 3(f). Word cloud is two words in line 19 but is one word in Title of Figure 2 and possibly elsewhere.

4.) The Reference s on pages 10 and 11 need very carefully editing. I noticed that References 8, 15, 19, 20, 21, 25, 27, 28 and 29 use capital letters for each of the words of the Titles but the other References do not. This is illustrates inconsistent formats used for presentation of the References.

Author Response

Point 1: Is CONCUR an acronym? If so, it needs to be spelled out in first occurrence in Abstract line 20. CONCUR analysis is discussed in line 120 but no explanation or presentation of formula used to perform these calculations mentioned on lines 120 and 121. CONCUR Analysis is further discussed in section 3.4 but no formula(s) for the calculations discussed in line 120 are presented in this section 3.4.

Response 1: The following content has been revised to reflect the reviewer's comments.

Finally, CONCOR analysis uses Python to calculate the frequency of simultaneous ap-pearance of the top 100 words in each paragraph, and converts the word list into a one-mode symmetric matrix (word × word) using the Ucinet 6.0 ver. program. The Net-Draw program then analyzed the correlation and visualized and clustered the network between words)

Point 2: This paper presents visual presentations of the techniques of word cloud, n-gram, two-way word tree and CONCUR analysis but do not provide any of the calculations referred to in the descriptions of CONCUR Analysis as it does for the others as in sections 3.1, 3.2 and 3.3.

Response 2: The contents of the analysis process were added and modified.

Point 3: Table 1 is used twice for "Wordcloud Analysis Results" on page 4 and also for N-gram Analysis Results on page 6 that instead should be Table 2. There are no figure numbers for the 6 figures on page 7 that maybe should be labeled as Figures 3(a), 3(b), 3(c), 3(d), 3(e) and 3(f). Word cloud is two words in line 19 but is one word in Title of Figure 2 and possibly elsewhere.

Response 3: I modified it as you requested.

Point 4: The Reference s on pages 10 and 11 need very carefully editing. I noticed that References 8, 15, 19, 20, 21, 25, 27, 28 and 29 use capital letters for each of the words of the Titles but the other References do not. This is illustrates inconsistent formats used for presentation of the References.

Response 4: It has been modified to match the reference form required by the journal.
